# Effects of Lead and Zinc Exposure on Uptake and Exudation Levels, Chlorophyll-a, and Phycobiliproteins in *Sarcodia suiae*

**DOI:** 10.3390/ijerph20042821

**Published:** 2023-02-05

**Authors:** Chia-Ching Chang, Chung-Chih Tseng, Tai-Wei Han, Beta Susanto Barus, Jhih-Yang Chuech, Sha-Yen Cheng

**Affiliations:** 1Department of Dentistry, Zuoying Branch of Kaohsiung Armed Forces General Hospital, Kaohsiung City 81342, Taiwan; 2Zuoying Branch of Kaohsiung Armed Forces General Hospital, Kaohsiung City 81342, Taiwan; 3Institute of Medical Science and Technology, National Sun Yat-sen University, Kaohsiung City 80424, Taiwan; 4Department of Environmental Biology and Fisheries Science, National Taiwan Ocean University, Keelung 20224, Taiwan; 5Department of Marine Science, Faculty of Mathematic and Natural Science, Sriwijaya University, Indralaya 30662, Indonesia; 6Center of Excellence for Ocean Engineering, National Taiwan Ocean University, Keelung 20224, Taiwan

**Keywords:** *Sarcodia suiae*, uptake, exudation, phycobiliprotein, chlorophyll-a

## Abstract

The present study aimed to determine the changes in the biosorption, bioaccumulation, chlorophyll-a (chl-a), phycobiliproteins, and exudation in the red seaweed *Sarcodia suiae* exposed to lead and zinc. The seaweed was exposed to ambient lead and zinc environments for 5 days before being transferred to fresh seawater, and the changes in biodesorption, biodecumulation, chl-a, and phycobiliprotein levels in *S. suiae* were investigated. Lead and zinc biosorption and bioaccumulation in the seaweed increased with the increase in the lead and zinc concentrations and exposure times. Meanwhile, the biosorption and bioaccumulation of zinc in the seaweed following exposure to zinc were significantly higher (*p* < 0.05) than the biosorption and bioaccumulation of lead in the seaweed following exposure to lead with the same concentration at each exposure time. The chl-a, phycoerythrin (PE), phycocyanin (PC), and allophycocyanin (APC) contents in the seaweed significantly decreased with the increase in the lead and zinc concentrations and exposure times. The chl-a, PE, PC, and APC contents in *S. suiae*, *which was* exposed to 5 Pb^2+^ mg/L for 5 days, were significantly higher (*p* < 0.05) than those in the seaweed exposed to zinc at the same concentration and for the same exposure times. In the lead and zinc exudation tests, the highest biodesorption and biodecumulation were observed on the 1st day of exudation after the seaweed was transferred to fresh seawater. The residual percentages of the lead and zinc in the seaweed cells were 15.86% and 73.08% after 5 days of exudation, respectively. The biodesorption rate and biodecumulation rate of the seaweed exposed to lead were higher than those of the seaweed exposed to zinc. However, the effect of lead on chl-a and phycobiliproteins was greater than that of zinc. This might be the result of lead not being a necessary metal for these algae, whereas zinc is.

## 1. Introduction

Rapid population growth and industrial development have led to an increase in pollutants in aquatic systems [1,2]. Among pollutants, heavy metal contamination is one of the primary pollution sources. Coastal areas are particularly vulnerable to serious environmental problems caused by artificial pollutants [3]. Heavy metals pose a severe hazard to aquatic animals, intertidal organisms, and humans in coastal environments. In coastal ecosystems, heavy metals are present in a dissolved state in water, or they are deposited in the sediment according to different environmental factors (such as pH, salinity, and conductivity) [4,5]. Lead, copper, cadmium, zinc, and nickel are the most common heavy metal pollutants [6].

Photosynthesis is the primary physiological function of algae growth. Chlorophyll is the major photosynthetic pigment closely related to plant photosynthesis. Reduced chlorophyll is an important indicator of plant aging [1]. Phycobiliproteins, such as phycoerythrin (PE), phycocyanin (PC), and allophycocyanin (APC), are water-soluble proteins composed of covalently bound proteins and phycobilisomes, and they are commonly found in red seaweed, blue-green algae, crypto algae, and some dinoflagellates. Algal photosynthesis and environmental adaptability are connected to the amounts of these pigments. In the body of *Nostoc sphaeroides*, measurement experiments have revealed that PC and APC increase with light intensity, whereas PE decreases. In photosynthesis, phycobilisomes transmit the wavelengths required for light reactions to chlorophyll in order to compensate for the lack of chlorophyll light energy absorption. These phycobilisomes help organisms by significantly increasing their capacity for absorbance by absorbing a large portion of the visible spectrum. They extend into the stromal area and are closely connected to the photosynthetic membrane. Cross-sections of the photosynthetic membrane can be seen with electron microscopy, as can the membrane surface when viewed from the stromal side [7,8,9].

Heavy metals affect algal photosynthetic pigments and growth [10,11,12]. The toxic effects of heavy metals on plants are reflected in the inhibition of photosynthesis and nitrogen fixation, reduced enzyme activity, and impact on algal metabolism and physiological processes [1,9]. Heavy metal ions also affect the stability of phycobiliproteins and the transfer of electrons in photosynthesis, inhibiting the growth and causing the death of algae [13]. Han et al. [12] indicated that cadmium affected the bioaccumulation, biosorption, and photosynthesis of *Sarcodia suiae*. Moreover, chl-a and phycobiliproteins were also decreased by cadmium.

Algae and seaweed are indispensable parts of the ecosystem in coastal areas. Seaweed accounts for a high proportion of the coastline. However, coastal waters are most affected by anthropogenic pollutants. Therefore, the heavy metal biosorption and bioaccumulation of seaweed can reflect the impact of heavy metal pollution along the coast [9]. *Sarcodia suiae* is one of the red seaweeds wildly distributed in the intertidal and subtidal zones throughout Indo–western Pacific regions [14]. *S. suiae* is a macroalgae with a high price, and it contains various nutrients, including protective pigments, antioxidants in the form of certain vitamins (A, C, E, and B12), and trace minerals, which are crucial for the health and performance of the thyroid. Besides being used in the food industry, red seaweed is also an important ingredient in the cosmetic industry. Recently, it has been widely cultivated in Japan and Taiwan as a healthy food.

Previous studies on the effect of algae on heavy metal uptake have mainly focused on microalgae [15], brown algae [16], and green algae [17]. Red seaweed is one of the most crucial marine macro-algae with a high economic value. The coastal environment inhabited by red seaweed is suffering from severe artificial pollution, including heavy metal pollution [6], but there is little research to study this. Therefore, this study investigated the heavy metal uptake and exudation of the seaweed *Sarcodia suiae* to determine the effects on and changes in chl-a and phycobiliproteins in seaweed exposed to the heavy metals lead and zinc. This study used zinc and lead as heavy metals to compare the effects of essential and non-essential heavy metals on the pigments present in *S. suiae*. The changes in the above parameters during the heavy metal exudation experiment were also observed.

## 2. Materials and Methods

Seawater was pumped from the Keelung coast adjacent to the National Taiwan Ocean University. The extracted seawater was precipitated and filtered with sand and gravel. Furthermore, the seawater was filtered using a nanostructured cellulose sponge filter to decontaminate pollutant materials, such as heavy metals and organic materials. The water was stored in a 2 tons FRP bucket in a greenhouse, and it was used after full aeration for 3 days. Before being used, it was ensured that the seawater was not contaminated with heavy metals so that it had no effect on the seaweed. 

Lead and zinc test solutions were prepared by dissolving Pb(NO_3_)_2_ and Zn(NO_3_)_2_•6H_2_O (Merck, Darmstadt, Germany) in 1 L of distilled water to make 1000 mg/L lead and zinc stock solutions [18], which were then diluted to 1 and 5 mg/L, respectively. This concentration was chosen because the experiment was carried out in a short time period, so a high concentration was needed to see the effect of the exposure.

*Sarcodia suiae*, a red seaweed, was bought from Pingdong County in Taiwan (5.04 ± 0.25 g). Before testing, the seaweed was adapted in fiberglass-reinforced plastic tanks for two weeks at a salinity of 34‰. The water was kept at a constant 20–25 °C temperature. The dissolved oxygen (DO) and pH levels were kept at 7.86 ± 0.74 mg/L and 8.33 ± 0.05, respectively. Ammonium nitrate (NH_4_NO_3_, 160 mg/L) and water-soluble phosphorus (P_2_O_5_, 4000 mg/L) were provided as nutrients to the saltwater during the acclimatization phase [12]. There was no discernible difference in the mean (±SD) wet body weight between treatments (*p* > 0.05).

We use a 6 L plastic white tank filled with 1 L of experimental solution. One piece of weighted seaweed was placed in each tank, which was then capped and placed in a water bath (22 ± 1 °C). The seawater salinity and pH level were 35‰ and 8.32 ± 0.06, respectively. The light intensity was 55 μmol photon/m^2^/s under a photoperiod for 24 h [8]. The experimental water was changed every 24 h, and nutrient salts were added. The experiment was divided into 3 groups based on the concentration of the heavy metals that the seaweed was exposed to: 0 (control), 1, and 5 mg/L of heavy metals. The adapted seaweed was placed in fresh seawater that was not contaminated with heavy metals as a control. This control was used to analyze the chlorophyll-a content. There were six replicates for each experimental group, and the experimental times were 0, 1, 2, and 5 days.

On a flame atomic absorption spectrometer (SpectrAA 240FS, Varian, Palo Alto, CA, USA), the test solutions’ lead and zinc concentrations were assessed. Licata et al. [19] provide illustrations of the experimental setup and the accuracy and precision of the analytical approach. The experimental setup of this study is illustrated in Figure 1 below.

The lead and zinc that the algal surface had absorbed were measured after washing the seaweed samples with HCl 0.1 mol/L for 30 s. The lead and zinc in the algal body were then detected using the seaweed samples. The seaweed samples that were acid washed were removed right away and frozen at −20 °C. After that, the samples were dried at −60 °C using a freeze-dryer (FD-20A2D, H.C.S., Taipei, Taiwan). After three days, the samples were treated with strong nitric acid (70%), which was then digested in a microwave-accelerated reactor (MARS Xpress, CEM, Matthews, NC, USA) [12]. The Pb^2+^ and Zn^2+^ ions in the seaweed were assessed using the flame atomic absorption spectrophotometer.

We calculated the biosorption and bioaccumulation with the seaweed wet weight using the following formula [12]:$$Biosorption(\mu g/g)=\frac{C_{1}\times V_{1}}{W_{wet}}$$
$$Bioaccumulation(\mu g/g)=\frac{C_{2}\times V_{2}}{W_{wet}}$$

*C*_1_: heavy metal concentration from seaweed’s surface acid-wash solution by AA;

*C*_2_: heavy metal concentration from seaweed’s microwave-digested solution by AA;

*V*_1_: sample volume for biosorption (10 mL);

*V*_2_: sample volume for bioaccumulation (10 mL);

*W_wet_*: wet weight (g).

The samples of dried seaweed were weighed, and then distilled water was added. Using a model FastPrep-24 5G (MP Biomedicals, Santa Ana, CA, USA), the samples were homogenized twice at 13 °C (Thermo Fisher Scientific, Waltham, MA, USA), and then they were centrifuged at 3000× *g* for 5 min at 4 ℃ with a Model 5403 centrifuge (Thermo Fisher Scientific, Waltham, MA, USA). The supernatant was taken off, 90% acetone was added, and the mixture was incubated at 4 °C for two hours. A spectrophotometer (GE Healthcare Life Science, Buckinghamshire, UK) was used to measure the absorbance at 630, 647, 664, and 750 nm. Utilizing the acetone volume and the seaweed wet weight, the following formula was applied to determine the concentration of chl-a [20]:$$Chl-a\;(\mu g)=11.85\times ({Abs}_{644}+{Abs}_{750})-1.54\times \left({Abs}_{647}-{Abs}_{750}\right)-0.05\times ({Abs}_{630}-{Abs}_{750})$$

Phosphate-buffered saline (PBS) was added after weighting the dried seaweed samples. The samples were homogenized twice at 13 °C using a model FastPrep-24 5G from MP Biomedicals in Santa Ana, California, and then they were centrifuged at 3000× *g* for 5 min at 4 °C using a Model 5403 centrifuge (Thermo Fish Scientific, Waltham, MA, USA). Using an ELISA reader (VersaMax Microplate Reader, Molecular Devices, Silicon Valley, San Francisco, CA, USA), we measured the absorbance of the supernatant at wavelengths of 562, 615, and 652. Using the seaweed wet weight and the following formula, we determined the PE, PC, and APC contents [21]:$$\mathrm{PC}\;(\mathrm{mg})=\frac{{Abs}_{615}-(0.474\times {Abs}_{652})}{5.35}$$
$$APC(\mathrm{mg})=\frac{{Abs}_{652}-(0.208\times {Abs}_{615})}{5.09}$$
$$PE(mg)=\frac{{Abs}_{562}-\left(2.41\times PC\right)-(0.849\times APC)}{9.62}$$

In the exudation experiment, we analyzed the biodesorption and biodecumulation of the heavy metals in the seaweed. Six replicates of the seaweed were exposed to 5 mg/L lead and zinc environments following the above exposure test for 5 days for each heavy metal concentration. The lead and zinc concentrations in the water and the seaweed were measured by replacing the seaweed with fresh seawater and sampling every 1, 2, and 5 days. The seaweed sample was taken to analyze chl-a and phycobiliproteins using the same protocol described above. We calculated the biodesorption and biodecumulation with the seaweed wet weight using the following formula:$$Biodesorption(\mu g/g)=\frac{E_{1}\times V_{3}}{W_{wet}}$$
$$Biodecumulation(\mu g/g)=\frac{E_{2}\times V_{4}}{W_{wet}}$$

*E*_1_: heavy metal concentration from seaweed’s surface acid wash solution by AA;

*E*_2_: heavy metal concentration from seaweed’s microwave-digested solution by AA;

*V*_3_: sample volume for biodesorption (10 mL);

*V_4_*: sample volume for biodecumulation (10 mL);

*W_wet_*: wet weight (g).

One- and two-way ANOVAs were used to analyze all the data. Duncan’s multiple-range test was performed to determine whether there were significant differences between treatments if a difference was shown at the 0.05 level. All tests’ statistical significance was accepted at the *p* < 0.05 level.

## 3. Results

The heavy metal biosorption and bioaccumulation levels of *S. suiae* significantly increased with the increase in the lead and zinc concentrations and exposure times (*p* < 0.05). The lead biosorption and bioaccumulation of the seaweed exposed to 5 Pb^2+^ mg/L for 5 days were 67.832 ± 15.036 and 197.380 ± 31.022 μg/g, respectively (Figure 2A). At the same exposure time with 5 Zn^2+^ mg/L for 5 days, the zinc biosorption and bioaccumulation of the seaweed were 869.533 ± 75.500 and 1353.931 ± 181.449 μg/g, respectively (Figure 3A). The ratios of biosorption to bioaccumulation were 0.2474, 0.3005, and 0.3437 for the seaweed exposed to 5 Pb^2+^ mg/L for 1, 2, and 5 days, and these ratios were 0.8486, 0.6984, and 0.6422 for 5 Zn^2+^ mg/L for the same exposure times, respectively.

The chl-a content was 0.610 ± 0.025 mg/g in the control, and it significantly decreased with the increase in the ambient lead and zinc concentrations and exposure times (*p* < 0.05). *S. suiae was* exposed to 1 and 5 Pb^2+^ mg/L for 5 days, and the chl-a contents were 0.361 ± 0.023 and 0.251 ± 0.024 mg/g, respectively. There were significant differences in each exposure time (*p* < 0.05). *S. suiae* was exposed to 1 and 5 Zn^2+^ mg/L for 5 days, and the chl-a contents were 0.552 ± 0.009 and 0.331 ± 0.023 mg/g, respectively (Figure 4). There were significant differences in each exposure time (*p* < 0.05). The seaweed was exposed to 5 mg/L lead and zinc for 5 days, and the chl-a decreasing percentage was 41.15% in 5 mg/L lead and 54.26% in 5 mg/L zinc (Figure 3). 

In the control, the phycobiliprotein content (summary of PE, PC, and APC) was 8.658 ± 0.470 mg/g. The phycobiliprotein contents were 2.489 ± 0.093, 1.443 ± 0.050, and 1.182 ± 0.059 mg/g when the seaweed was exposed to 5 Pb^2+^ mg/L for 1, 2, and 5 days, respectively. The phycobiliprotein contents were 6.901 ± 0.393, 5.501 ± 0.337, and 3.834 ± 0.169 mg/g when the seaweed was exposed to 5 Zn^2+^ mg/L for 1, 2, and 5 days, respectively (Figure 5A).

The PE, PC, and APC contents were 3.513 ± 0.278, 3.331 ± 0.354, and 1.814 ± 0.138 mg/g in the control, respectively. The PE, PC, and APC contents decreased to 0.383 ± 0.040, 0.442 ± 0.049, and 0.357 ± 0.031 mg/g in 5 Pb^2+^ mg/L after 5 days of exposure, respectively. There were also significant differences with the control (*p* < 0.05). The PE, PC, and APC contents decreased to 1.414 ± 0.148, 1.746 ± 0.014, and 0.674 ± 0.077 mg/g in 5 Zn^2+^ mg/L after 5 days of exposure, respectively. Moreover, there was also a significant difference with the control (*p* < 0.05). *S. suiae* samples were exposed to 5 Pb^2+^ mg/L for 5 days, and the PE, PC, and APC decreasing percentages were 89.10%, 86.73%, and 80.32% compared with those of the control, respectively. When the seaweed was exposed to 5 Zn^2+^ mg/L for 5 days, the PE, PC, and APC decreasing percentages were 59.75%, 47.58%, and 62.85%, respectively. The ratios of PE, PC, and APC to phycobiliproteins of *S. suiae* were 40.58%, 38.47%, and 20.95% in the control, respectively. In the *S. suiae* seaweed exposed to 5 Pb^2+^ mg/L for 5 days, the ratios of PE, PC, and APC to phycobiliproteins were 32.40%, 37.39%, and 30.20%, respectively. However, in the seaweed exposed to zinc, the ratios of PE, PC, and APC to phycobiliproteins were 36.88%, 45.54%, and 17.58%, respectively (Figure 5A).

After the seaweed was exposed to 1 and 5 Pb^2+^ mg/L for 5 days, it was then transferred to fresh seawater. The biodesorption rate and biodecumulation rate were increased with the increase in exudation time, and there were significant differences with the initial lead exudation (*p* < 0.05). At 1 Pb^2+^ mg/L for 5 days of exudation, the seaweed biodesorption rate and biodecumulation rate were 94.97% and 88.91%, respectively. Moreover, those at 5 Pb^2+^ mg/L for 5 days of exudation were 92.72% and 84.14%, respectively (Figure 2B). The biodesorption rate and biodecumulation rate in the zinc exudation test also increased with the increase in exudation time, and they were significantly lower than those in the lead exudation test (*p* < 0.05). The biodesorption rate and biodecumulation rate were 31.05% and 16.76% in 1 Zn^2+^ mg/L for 5 days of exudation, and they were 48.27% and 26.92% in 5 Zn^2+^ mg/L for 5 days of exudation, respectively. Moreover, there were significant differences with the initial zinc exudation (*p* < 0.05) (Figure 3B).

In the 5 mg/L lead and zinc exudation experiment, after 5 days, the residual percentages of lead on the seaweed surface and inside the seaweed were 7.28% and 15.86%, respectively. However, the residue percentages of zinc on the seaweed surface and inside the seaweed were 51.73% and 73.08%, respectively.

On the fifth day of exudation, the chl-a levels in the seaweed exposed to lead and zinc were 0.242 ± 0.022 and 0.352 ± 0.023 mg/g, respectively. After 5 days of lead and zinc exudation, the chl-a contents recovered to 0.490 ± 0.041 and 0.394 ± 0.043 mg/g, respectively. The chl-a recovery percentages were 81.66 and 62.60% after 5 days of lead and zinc exudation, respectively (Figure 4B).

The phycobiliprotein contents were 5.100 ± 0.286, 5.606 ± 0.243, and 6.595 ± 0.236 mg/g in the 5 Pb^2+^ mg/L exudation experiments lasting for 1, 2, and 5 days, respectively. The phycobiliprotein contents were 3.916 ± 0.173, 4.557 ± 0.230, and 5.020 ± 0.165 mg/g in the 5 Zn^2+^ mg/L exudation experiments lasting for 1, 2, and 5 days, respectively (Figure 5B).

In the seaweed exposed to 5 mg/L lead and zinc separately for 5 days, the PE contents were 0.363 ± 0.281 and 1.408 ± 0.036 mg/g, the PC contents were 0.451 ± 0.032 and 1.735 ± 0.209 mg/g, and the APC contents were 0.337 ± 0.041 and 0.664 ± 0.075 mg/g after the initial exudation day, respectively. In the 5 Pb^2+^ mg/L exudation experiment, the PE, PC, and APC contents recovered to 2.948 ± 0.178, 2.061 ± 0.208, and 1.586 ± 0.150 mg/g after 5 days of exudation, respectively. The recovery percentages of PE, PC, and APC in 5 Pb^2+^ mg/L were 83.92%, 61.86%, and 87.45% after 5 days of exudation, respectively. In the 5 Zn^2+^ mg/L exudation experiment, the PE, PC, and APC contents recovered to 2.100 ± 0.037, 1.954 ± 0.147, and 0.965 ± 0.082 mg/g after 5 days of exudation, respectively. The recovery percentages of PE, PC, and APC were 59.79%, 58.67%, and 53.22% after 5 days of exudation, respectively. The recovery percentages of PE, PC, and APC in the zinc exudation experiment were significantly lower than those in the lead exudation experiment (*p* < 0.05) (Figure 6B).

The heavy metal content in the seaweed had a negative correlation with the chl-a content in the seaweed. The lead content in the seaweed had a more significant impact on chl-a than zinc (Figure 7A). In the exposure and exudation experiments, the phycobiliprotein content showed a positive correlation with the chl-a content in the seaweed. The effect of lead on phycobiliproteins was greater than that of zinc at the same chl-a content (Figure 7B).

## 4. Discussion

Algae’s heavy metal absorption abilities are related to their cell wall structures and physiological characteristics. Their inner layers are cellulose, and their surfaces are wrinkled to increase their adsorption surface areas [22]. Polymers, such as polysaccharides, proteins, and phospholipids, on the cell wall provide algae with functional groups that bind to heavy metal ions [23]. Different species of algae have different cell wall compositions, resulting in differences in their biosorption capacities for heavy metal ions. For example, alginate in brown algae cell walls has a high affinity for lead ion absorption [24]. Red seaweed has a high biosorption capacity for lead ions. The cell wall is mainly composed of carrageenan and agar [25,26]. This present study found that *S. suiae* exposed to 5 mg/L of lead for 1 and 5 days had a biosorption of 10.964 ± 2.817 and 67.832 ± 15.036 μg/g, respectively. The *S. suiae* samples were exposed to 5 mg/L zinc for 1 and 5 days, and the bioabsorption was 185.453 ± 7.349 and 869.533 ± 75.503 μg/g, respectively. These results indicate that red seaweed *S. suiae* has high biosorption capacities for lead and zinc ions. 

Chen and Han [27] demonstrated heavy metal biouptake, including biosorption and bioaccumulation. Brinza et al. [28] showed that *Fusus vesiculosus* is an effective biosorbent for absorbing zinc ions from sewage, reducing the concentration of heavy metals in water. Doshi et al. [29] also mentioned that *Laminaria hyperborea*, *Sargassum muticum*, and *Fucus spiralis* have been shown to be effective in removing heavy metals (Cd^2+^, Zn^2+^, and Pb^2+^). In this present study, we also found that the amount of heavy metals absorbed on the surface of the seaweed and accumulated in its interior increased with the increase in exposure time and the concentrations of the heavy metals. The bioaccumulation of lead and zinc in *S. suiae* reached 44.30 ± 2.09 and 218.53 ± 23.81 µg/g after exposure to 5 mg/L of lead and zinc for 1 day, respectively; the bioaccumulations were 197.380 ± 31.022 and 1353.931 ± 181.449 µg/g after exposure for 5 days. These results indicate that *S. suiae* not only adsorbs heavy metals, such as lead and zinc, but that it also accumulates them. For both lead and zinc, the bioaccumulation was higher than the biosorption. Meanwhile, the biosorption and bioaccumulation of lead were significantly higher than those of lead.

Lead is the most common and prevalent heavy metal in the environment, with adverse effects on algae and phytoplankton [30,31,32]. Lead can be phytotoxic, even at low concentrations [33]. Lead ions induce toxicity in plants through an ionic mechanism. The toxicity mechanism of lead ions is mainly due to the ability of lead ions to replace other divalent cations, such as Ca^2+^, Mg^2+^, and Fe^2+^, and monovalent cations (Na^+^), which, in turn, interferes with cellular metabolic functions, causing significant changes in intracellular and intercellular signaling, protein folding, maturation, and apoptosis [34].

Bajguz found that heavy metals impair *Chlorella vulgaris* growth [35]. Furthermore, Bajguz [36] confirmed that heavy metals affect the reduction of significant substances in cells, such as enzymes and photosynthetic dyes, and that they change the protein content, sugars, or fats in algae cells, as well as the intensity of key metabolic processes related to the transport of vital cells and ions from cellular sites, such as the process of photosynthesis. Lead is not an essential element for growth, and plants actively exclude or sequester non-essential elements to minimize and avoid the toxicity of heavy metals. Zinc is an essential element for living organisms [37]. Ralph and Burchett and Gabryś et al. [38,39] mentioned that metabolic processes (including photosynthesis) also require copper and zinc. Zinc is involved in the regulation of sugar metabolism and protein synthesis. Plants actively absorb these elements. A high concentration of zinc inhibits the growth of algae, affects photosynthesis, and reduces the content of chlorophyll, resulting in an imbalance in the ratio of carotenoids to chlorophyll. Other studies have also shown that zinc can reduce the photosynthetic capacity of phytoplankton, even at low concentrations. *Chlorella vulgaris* has been found to show a 50% reduction in its growth rate when exposed to a zinc concentration of 2.4 ppm, while *C. vulgaris* growth has been found to stop completely when exposed to 20 ppm [40]. The results of this study show that lead enters the seaweed and accumulates inside it directly. The percentages of bioaccumulation to biouptake were 68.24–80.16% following lead exposure. However, the percentages of bioaccumulation to biouptake were 50.20–60.89% following zinc exposure. Although both the biosorption and bioaccumulation of lead and zinc increased with the increase in the exposure concentrations and exposure times, the biosorption and bioaccumulation of zinc were significantly higher than those of lead. 

Algae’s biouptake of heavy metals includes rapid extracellular absorption and slow intracellular accumulation; Rangsayatom et al. [41] called these the rapid phase and the slower phase, respectively. In this present study, the seaweed was exposed to lead and zinc environments. The biosorption and bioaccumulation of zinc in the seaweed were significantly higher than those of lead. It is speculated that the possible reason for this is that zinc is an essential heavy metal, but lead is not. Previous studies have found that *H. ovalis* can uptake zinc and that it is more tolerant to zinc than other heavy metals [42]. Woolhouse [43] indicated that the competition between zinc and iron during chlorophyll biosynthesis, which requires iron ions, may be due to the loss of pigment, while plants can still tolerate increased zinc ion concentrations. *H. ovalis* may produce phytochelalins that bind zinc ions, providing a mechanism for the ability to uptake zinc.

Diannelidis and Delivopoulos [44] found that, in the macroalga *Ceramium ciliatum* exposed to cadmium, thylakoids were destroyed, and plastid spheres increased in the chloroplast. In red seaweed, phycoerythrin is located in the outer layer region of the phycobilisome, making it more susceptible to environmental factors [8]. High concentrations of lead have been found to disrupt protein and thylakoid membranes, and they have also been found to cause the inhibition of enzymatic activities [45] and photosynthesis [46].

Phycobilisomes are composed of aggregates containing covalently bound phycobiliproteins. Phycobiliproteins include phycoerythrin (PE), phycocyanin (PC), and isophycocyanin (APC). Peripheral PE first absorbs the light energy, and the light energy absorbed by the PE is transferred to the PC, then to the APC, and, finally, to the photoreaction system II and photoreaction system I reaction centers [47].

This present study also found that chl-a and phycobiliproteins decreased with the increase in the ambient lead and zinc concentrations and exposure times. The decrease in phycobiliproteins observed when the seaweeds were exposed to lead and zinc may be due to the destruction of chl-a and phycobiliproteins by the heavy metal uptake. Dos Santos et al. [1] treated *G. domingensis* with cadmium and found that the phycobiliprotein content decreased. High concentrations of cadmium changed the structures of the phycobilisomes. Han et al. [12] also found similar results. In their study, *S. suiae* was exposed to cadmium for 24 h, and chl-a, phycoerythrin, phycocyanin, and isophycocyanin significantly decreased. The ratio of phycobiliproteins to chl-a increased with an increase in the exposure time. There was a more significant reduction in chl-a than in phycobiliproteins in the seaweed exposed to cadmium. Meanwhile, this present study also observed that the ratio of phycobiliproteins to chl-a decreased with the increase in heavy metal concentrations and exposure times.

A high lead concentration can reduce plant growth and affect physiologic development, resulting in reduced photosynthesis and electron transport rates [48,49,50]. Lead has been found to directly inhibit photosynthetic electron transport, enzymatic activity in the Calvin cycle, and net carbon dioxide usage [49]. Lead has also been found to indirectly alter photosynthetic activity, affect chloroplast ultrastructure, and reduce leaf chlorophyll content [51].

Gu and Lan [52] investigated the physiological and biochemical properties of lead-caused effects in the green alga *Neochloris oleoabundans*. Chlorophyll decreased with an increase in the lead concentration, and this decrease in chlorophyll may have been due to the inactivation of enzymes caused by the substitution reaction of magnesium with lead. Other reports have also found a decrease in chl-a content in the leaves of plants treated with lead [53]. Gouveia et al. [10] investigated the effects of lead and copper on cell structure and metabolism in red seaweed, and they found that lead and copper treatment significantly reduced the content of photosynthetic pigments in *Gracilaria domingensis* compared with those in control algae. The chl-a and phycobiliproteins contents were significantly reduced following lead and copper exposure. Some heavy metals, such as zinc and copper, although important cofactors involved in many biochemical processes (such as photosynthesis, the synthesis of reaction center proteins in PSII, and antioxidant systems), are toxic to algae at high concentrations [54].

Algal cells can reduce the entry of heavy metals via changes in membrane functions and changes in the cell wall [55,56]. This present study shows that lead is a non-essential heavy metal and that the seaweed cannot utilize this metal. At the end of the 5 Pb^2+^ mg/L exudation experiment, the biodesorption and biodecumulation of seaweed were only 7.28% and 15.86% after 5 days of exudation. On the contrary, the biosorption and bioaccumulation of zinc in the seaweed were higher than those of lead. The biodesorption and bioaccumulation were still 51.73% and 73.08% after 5 days of exudation. This result indicates the effects of and differences between essential and non-essential heavy metals in *S. suiae*.

In the exudation experiment results, the biodesorption rate and biodecumulation rate of lead were higher than those of zinc. Since lead is a non-essential heavy metal, the release of lead by the seaweed was significantly higher than that of zinc. It is also possible that the seaweed had less biosorption and bioaccumulation of lead during the exposure time. Therefore, lead ions could be eliminated within the exudation time, resulting in the biodesorption and biodecumulation of lead being higher than those of zinc. Zinc is an essential element in seaweed. The zinc content of the seaweed was too high during the exposure test, causing it to not be eliminated within the exudation time. Ralph and Burchett [36] showed that *Halophila ovalis* could accumulate zinc ions and that it is more tolerant to zinc than other heavy metals.

The chl-a content significantly recovered in the lead exudation experiment (*p* < 0.05). The chl-a recovery rates reached 93.04% and 81.66% in the 1 and 5 Pb^2+^ mg/L exudation experiments on the 5th day, respectively. However, in the exudation experiment of zinc, the chl-a contents also increased, but there were no significant differences. The chl-a recovery rates were only 93.54% and 62.60% in the 1 and 5 Zn^2+^ mg/L exudation experiments on the 5th day, respectively. It is speculated that the reason for this is the seaweed bioaccumulation of high concentrations of zinc. The residual zinc decreased the recovery rate of chl-a in the seaweed during the excretion time. The phycobiliproteins were also in the same condition. The chl-a and phycobiliproteins contents of the seaweed continually recovered after the seaweed eliminated the heavy metals. The recovery rates of chl-a and phycobiliproteins in the seaweed following lead exposure were higher than those following zinc exposure. These recovery rates had direct relationships with the residual heavy metals in the seaweed.

This present study shows the lead’s biodesorption rate and biodecumulation rate were higher than those of zinc. However, the effect of lead on the chl-a and phycobiliproteins in *S. suiae* was greater than that of zinc. On the contrary, zinc is an essential heavy metal for seaweed, and the seaweed accumulated high amounts of zinc. *S. suiae* could eliminate heavy metal ions after the uptake of lead and zinc. The physiological responses and changes in the chlorophyll-a and phycobiliproteins of *S. suiae* need further research.

## 5. Conclusions

The seaweed *S. suiae* was exposed to lead and zinc environments, and the biosorption and bioaccumulation of the seaweed increased with the increase in the heavy metal concentrations and exposure times. In addition, the chl-a and phycobiliproteins contents decreased with the increase in the heavy metal concentrations and exposure times. The biosorption and bioaccumulation of zinc in *S. suiae* following exposure to zinc were higher than the biosorption and bioaccumulation of lead in *S. suiae* following exposure to lead. However, the effect of lead on chl-a and phycobiliproteins was greater than that of zinc.

In the exudation experiment, the biodesorption and biodecumulation of the seaweed increased with the exudation time. This may be due to the lead being a non-essential heavy metal. On the contrary, zinc is an essential heavy metal for seaweed, and the seaweed accumulated high amounts of zinc. This study shows that exposure to heavy metals has an effect on the pigment content needed by *S. suiae*, which, in turn, affects its growth.

## Figures and Tables

**Figure 1 ijerph-20-02821-f001:**
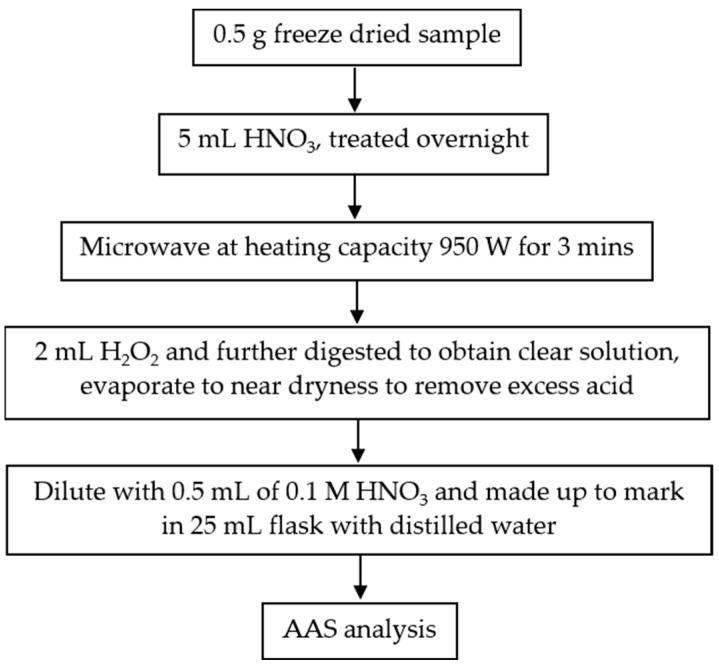
Experimental setup.

**Figure 2 ijerph-20-02821-f002:**
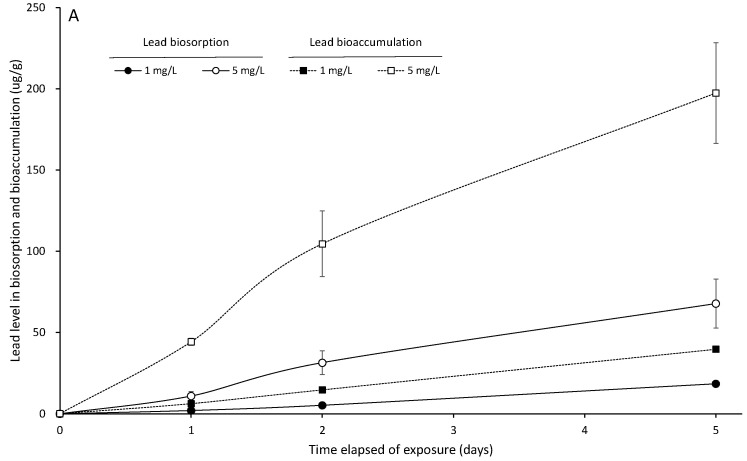
The lead biosorption and bioaccumulation (μg/g) of *S. suiae* exposed to 1 and 5 Pb^2+^ mg/L for 1, 2, and 5 days (**A**). The lead biodesorption and biodecumulation of *S. suiae* exposed to 1 and 5 Pb^2+^ mg/L for 1, 2, and 5 days (**B**).

**Figure 3 ijerph-20-02821-f003:**
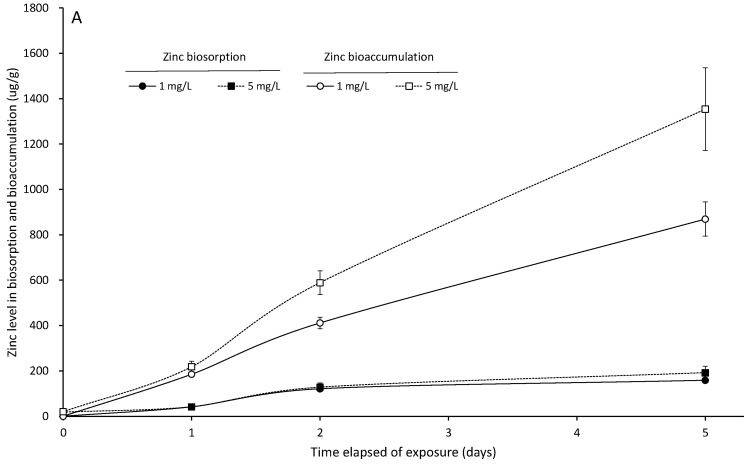
The zinc biosorption and bioaccumulation (μg/g) of *S. suiae* exposed to 1 and 5 Zn^2+^ mg/L for 1, 2, and 5 days (**A**). The zinc biodesorption and biodecumulation of *S. suiae* exposed to 1 and 5 Zn^2+^ mg/L for 1, 2, and 5 days (**B**).

**Figure 4 ijerph-20-02821-f004:**
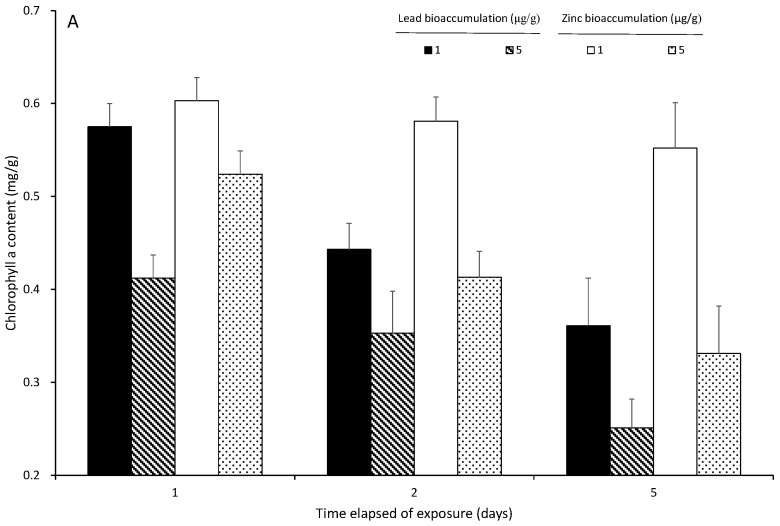
Chlorophyll-a contents (mg/g) of *S. suiae* exposed to various lead (1 and 5 Pb^2+^ mg/L) and zinc (1 and 5 Zn^2+^ mg/L) concentrations for 1, 2, and 5 days (**A**). Chlorophyll-a contents (mg/g) of *S. suiae* after 1, 2, and 5 days exudation following exposure to various lead (1 and 5 Pb^2+^ mg/L) and zinc (1 and 5 Zn^2+^ mg/L) concentrations for 5 days (**B**).

**Figure 5 ijerph-20-02821-f005:**
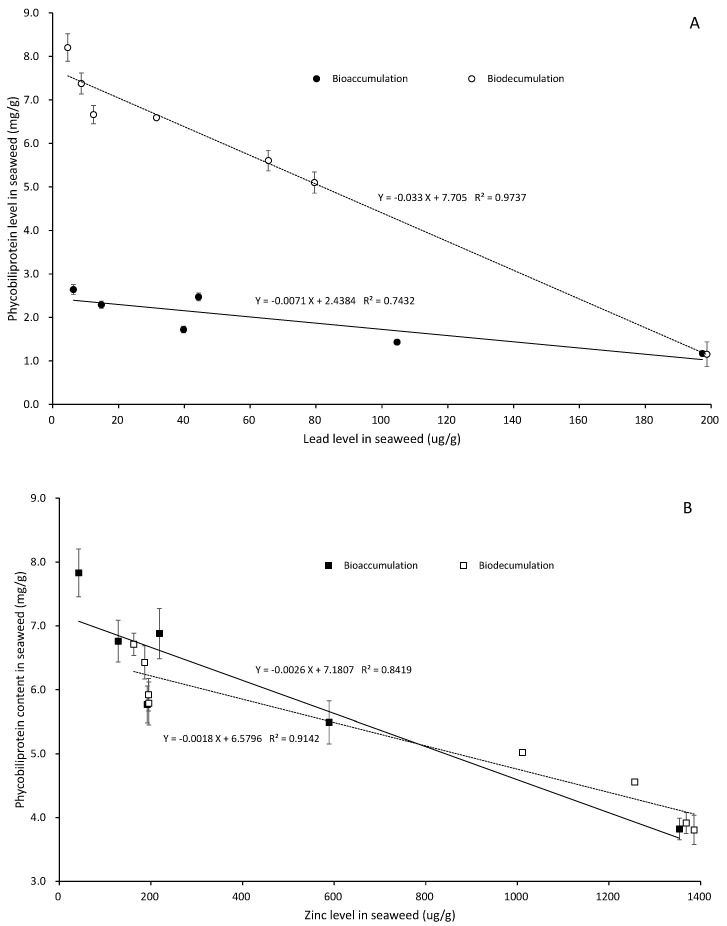
The relationship between phycobiliproteins and bioaccumulation and biodecumulation of lead (**A**) and zinc (**B**) in seaweed.

**Figure 6 ijerph-20-02821-f006:**
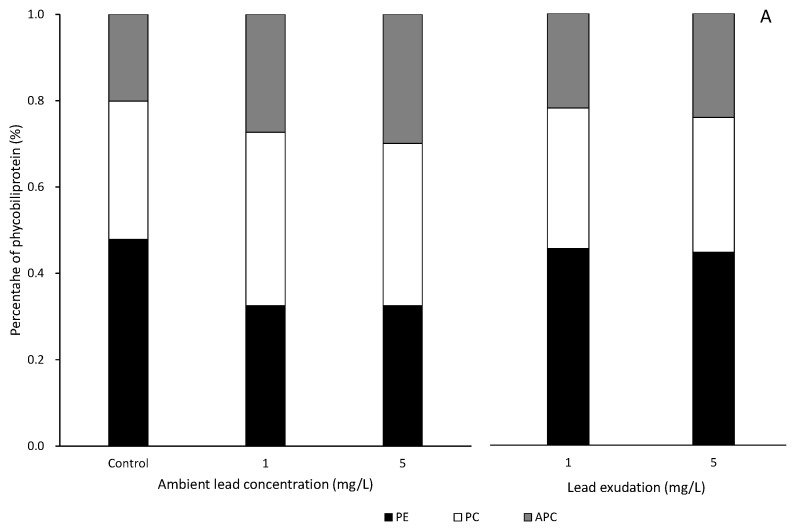
Proportional percentages of PE, PC, and APC to phycobiliproteins in *S. suiae* exposed to 0 (control), 1, and 5 Pb^2+^ mg/L at 5th day of exposure and 5th day of exudation (**A**). Proportional percentages of PE, PC, and APC to phycobiliproteins in *S. suiae* exposed to 0 (control), 1, and 5 Zn^2+^ mg/L at 5th day of exposure and 5th day of exudation (**B**).

**Figure 7 ijerph-20-02821-f007:**
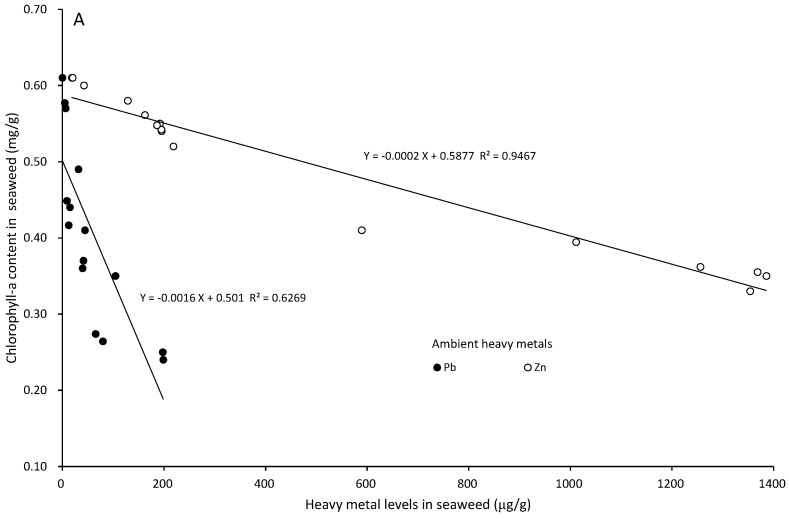
The relationship between chlorophyll-a (mg/g) and heavy metal levels (lead and zinc, µg/g) in *S. suiae* (**A**). The relationship between phycobiliproteins (mg/g) and chlorophyll-a (mg/g) in *S. suiae* (**B**).

## Data Availability

The raw data supporting the conclusions of this article will be made available by the authors without undue reservation.

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
