# Peer review of "Effects of Lead and Zinc Exposure on Uptake and Exudation Levels, Chlorophyll-a, and Phycobiliproteins in Sarcodia suiae"

_ijerph, 2023, doi:10.3390/ijerph20042821_

Round 1

Reviewer 1 Report (Previous Reviewer 1)

Most of the comments not addressed in this revised version. Abstract is again misleading in its last line. Methodology not upto the mark. Graphical setup of the experiment missing. Results are presented in chaotic manner. Discussion not much convincing. Though, reference style has been homogenised. No reference of 2021, 2022 

Author Response

Reviewer 2 Report (Previous Reviewer 2)

The manuscript titled "Effects of lead and zinc exposure on uptake and exudation levels, chlorophyll-a, and phycobiliprotein in Sarcodia suiae" was carefully reviewed. Most of the comments were resolved by the author and improve the standard of the manuscript. So I would like to recommend the journal editor accept this manuscript.

Author Response

Thank you for all your suggestions for this manuscript. We are very pleased and appreciate that you recommended the editor to be accepted for publication in this journal.

This manuscript is a resubmission of an earlier submission. The following is a list of the peer review reports and author responses from that submission.

Round 1

Reviewer 1 Report

The manuscript lacks hierarchy and coherence. Many repititions and confusing statements along the length of manuscript. Physicochemical parameters not taken into consideration to show their correlation with phycoerythrin (PE), phycocyanin (PC), and allophycocyanin (APC). Experimental setup (graphical representation/plate) is missing. No control experiment mentioned under the methodology section but mentioned under section of results and discussion. Moreover, no toxicological study has been done but it is concluded that lead is more toxic than zinc! There is a lot of scope for improvement. The authors are required to do major revision before its resubmission (pdf with comments enclosed).

Author Response

Dear Reviewer

We have revised the manuscript based on your comment. 

Best Regards,

Authors

Reviewer 1

First, we thank reviewers for their suggestions and comments on our articles. Comments and suggestions are very helpful for improving this manuscript. Here's our response to all your suggestions and comments.

This study investigated the heavy metal uptake and exudation on seaweed Sarcodia suiae to recognize the effects and changes of chl-a and phycobiliproteins on seaweed exposed to heavy metals lead and zinc. where there are not many studies that discuss this in detail. Here we want to show how the different effects caused by exposure to two different heavy metals namely lead and zinc on several parameters in seaweed. We have corrected some errors and also clarified some of the methods in this manuscript. In addition, we have also corrected and enhanced the language style of this manuscript through experts. The following is our response to comments from reviewers.

  1. What is biodecumulation??? It is again no scientific term

Response: Biodeccumulation is the opposite of bioaccumulation, which means the release of material from the body of red seaweed, in this case heavy metals. This term has also been used in other articles such as in articles with the following doi: https://doi.org/10.1016/j.scitotenv.2020.141197

  1. Why S. suiae was selected as a test plant?

Response:  The reason for choosing this species has been explained in the introduction (Lines 67-70)

  1. Is not there any threshold limit of uptake??/

Response: As far as we know, there is no reference that explains the limit of uptake.

  1. How many times the experiment was repeated?

Response: There were six replicates for each experimental group, and the experimental times were 0, 1, 2, and 5 days (Introduction 103-104).

  1. Artificial pollutants means???

Response: Pollution caused by manmade activities like deforestation, urbanization, industrialization, etc.

  1. . but had zinc and lead correlated with such factors in the present study??

Response: This statement is to convey that the concentration of heavy metals in the sea varies depending on these factors. In the present study we also used different concentrations of heavy metals.

  1. Roles of PC and APC not mentioned

Response: We have added the role of PC and APC in introduction.

  1. It is pertinent to mention that had photosynthetic activity in relation to study plant undertaken???

Response: We analyzed the chl-a, phycoerythrin (PE), phycocyanin (PC), and allophycocyanin (APC) contents in seaweed which are closely related to the process of photosynthesis.

  1. ....there is little study to study this... ????

Response: To our knowledge, not many studies have discussed this topic in detail

  1. The experiment is based on very duration. How many times the process has been repeated?

Response: The process of making stock is only done once which is then diluted to be used as needed in the experiment.

  1. Was the weed only found or taken for experiment??

Response: Seaweed was bought for experiment. We have revised it.

  1. acclimated???

Response: We have changed it with “adapted”.

  1. Then for what purpose seawater was pumped???

Response: Seawater is pumped for stock. Our university is close to the sea and experiments using seawater are obtained from the pumping results.

  1. The methodology is a bit confusing!! And no references taken... Is this the standard methodology?

Response: There are some mistake and we have revised it. This is the standart methodology and we design base on the condition of pumped seawater in our laboratory and describes the actual natural conditions.

  1. After freezing at -20, why and how were the samples again frozen?

Response: This is mistake and we have revised it.

  1. wet weight or fresh weight??

Response: We use the term wet weight following several related articles.

  1. what exudation experiment??? Setup is missing in methodology.

Response: Exudation in this study is biodesorption and biodecumulation. We've improved the sentences to be clearer.

  1. Statement looks like dictation.

Response: we have improve the statement.

  1. No pollution gradients taken into consideration. No control study undertaken

Response: We have revised and added about control in this study

  1. who have devised these equations????

Response: We design this formula from modifications to the biosorption and bioaccumulation formulas.

  1. was the photosynthetic activity assessed??

Response: We analyzed the chl-a, phycoerythrin (PE), phycocyanin (PC), and allophycocyanin (APC) contents in seaweed which are closely related to the process of photosynthesis.

  1. Does it mean adsorption and absorption rates were also studied???

Response: Yes, we did. Adsorption and absorption are included in bioaccumulation.

  1. How can it be justified?? Line 267

Response: This can be justified from Figure 6A where this figure shows that the correlation value (R2) for lead is higher than that for zinc.

  1. Phyto-toxicity not studied in this research attempt thus it is out of context to discuss it.

Response: We add this sentence to strengthen why the level of biodecumulation and biodesorption of lead is higher than zinc in seaweed. Lead is a heavy metal that is harmful to seaweed even in low concentrations, while zinc is needed by seaweed.

  1. No such parameters studied in this experiment. What is fun to discuss these????

Response: This paragraph explains how heavy metals have a negative effect on seaweed. This is still related to the parameters tested in this study. In addition, the statement can strengthen how important this study is.

  1. Repeating the same statement time as again...

Response: We have deleted it.

  1. Tartary buckwheat is not an aquatic/marine plant. Not justified

Response: We have deleted and revised it.

  1. Statement repeated again without valid justification

Response: All the statement described based on the results. We have added some information about it in the discussion.

  1. Recent references missing (2021 & 2022)

Response: We have added the recent reference in this manuscript.

Reviewer 2 Report

The manuscript titled "Effects of lead and zinc exposure on uptake and exudation of levels, chlorophyll-a, and phycobiliprotein in Sarcodia suiae" was carefully reviewed. However, there are a few conflicts to be resolved.

Please focus on English grammar and sentence formation before submitting the manuscript.

Please focus on the material and method sections. Please recheck the procedure once again. There are few flaws in experimental procedures. Please add references for the test carried out in this study.

Explain the heavy metal extraction process in detail.

Explain the assays with proper references.

Why did you choose only Sarcodia suiae? Was there any specific reason? If yes, please explain it in detail.

Did you maintain any control setup in this study?

The figures explained in the result section are not sufficient. A detailed explanation is needed.

In the discussion section, recent revelations and studies are to be discussed, along with current study outcomes.

The reference study followed. Check whether all the references are cited in the main text.

Author Response

Dear Reviewer

We have revised the manuscript based on your comment. Thank you.

Best Regards,

Authors

Reviewer 2

First, we thank reviewers for their suggestions and comments on our articles. Comments and suggestions are very helpful for improving this manuscript. Here's our response to all your suggestions and comments.

This study investigated the heavy metal uptake and exudation on seaweed Sarcodia suiae to recognize the effects and changes of chl-a and phycobiliproteins on seaweed exposed to heavy metals lead and zinc. where there are not many studies that discuss this in detail. Here we want to show how the different effects caused by exposure to two different heavy metals namely lead and zinc on several parameters in seaweed. We have corrected some errors and also clarified some of the methods in this manuscript. In addition, we have also corrected and enhanced the language style of this manuscript through experts. The following is our response to comments from reviewers.

Please focus on English grammar and sentence formation before submitting the manuscript.

Response: we have corrected and enhanced the language style of this manuscript through experts.

Please focus on the material and method sections. Please recheck the procedure once again. There are few flaws in experimental procedures. Please add references for the test carried out in this study.

Response: We have added some explanation and make the methods more clearly

Explain the heavy metal extraction process in detail.

Response: The heavy metal extraction process carried out in this study was an ordinary extraction as stated in the method. This method is also commonly described in previous articles

Explain the assays with proper references.

Response: We have added the supporting reference in the method, however, some assay was design by us regarding to real nature condition.

Why did you choose only Sarcodia suiae? Was there any specific reason? If yes, please explain it in detail.

Response: This species is used because it is widely cultivated in our country, has high economic value and is used as a healthy food ingredient as explained in the introduction.

Did you maintain any control setup in this study?

Response: Yes, we did. We already improve the method and added some information about control in the methods.

The figures explained in the result section are not sufficient. A detailed explanation is needed.

Response: We have added detailed explanation in the result

In the discussion section, recent revelations and studies are to be discussed, along with current study outcomes.

Response: We have added some information and recent reference in the discussion section.

Round 2

Reviewer 1 Report

·         There is so much scope for improvement, linguistically as well as technically.

·         After the second revision round, still major revision required.

·         Cannot be accepted in its present form.

·         After incorporation of major revision, it can be afresh resubmitted for consideration.

Author Response

First, we thank reviewers for their suggestions and comments on our articles. Comments and suggestions are very helpful for improving this manuscript. Here's our response to all your suggestions and comments.

This study investigated the heavy metal uptake and exudation on seaweed Sarcodia suiae to recognize the effects and changes of chl-a and phycobiliproteins on seaweed exposed to heavy metals lead and zinc. where there are not many studies that discuss this in detail. Here we want to show how the different effects caused by exposure to two different heavy metals namely lead and zinc on several parameters in seaweed. We have corrected some errors and also clarified some of the methods in this manuscript. In addition, we have also corrected and enhanced the language style of this manuscript through experts (MDPI English editing service). The following is our response to comments from reviewers.

Reviewer 2 Report

The manuscript titled "Effects of lead and zinc exposure on uptake and exudation of levels, chlorophyll-a, and phycobiliprotein in Sarcodia suiae" was carefully reviewed. However, there are a few conflicts to be resolved.

Still few typographic, spelling, and grammatical mistakes in the paper. Significant attention must be given to formatting, structure, and grammar.

Add some more important information about Sarcodia suiae in the introduction, including why you chose this seaweed. What are the commercial or industrial uses of the selected seaweed?

Please make use of the constant word samples or materials.

The control setup is necessary for the chlorophyll-a analysis.

Please check the sentence, "Seaweed samples that had been acid washed were removed right away and frozen at -20 °C. After that, samples were dried at -60 °C using a 121 freeze-dryer (FD-20A2D, H.C.S., Taipei, Taiwan)." It’s confusing; please clarify it. There are many sentences like this; please write them clearly.

Author Response

(The authors gave the same response as above.)
